# Foam film vitrification for cryo-EM

Yue Zhang [1] ✉, Biplob Nandy [1], Kasim Sader[2], Christopher J. Russo [1] ✉ & Jan Löwe [1] ✉

Electron cryomicroscopy (cryo-EM) has revolutionised structural biology, enhancing applicability, size limits and speed. Despite these successes, cryo-EM sample preparation remains a major bottleneck for routinely achieving high-resolution structures through single particle analysis. Challenges such as inconsistent ice thicknesses, air-water interface interactions and preferred particle orientation persist. Here, we introduce a blot-free vitrification method that uses free-standing surfactant-stabilised foam films to address some of these issues. The method achieves uniform ice thicknesses, enables thickness control of the foam film prior to vitrification, and for some specimens enhances orientation distribution efficiency. Furthermore, it reduces particle adsorption to carbon foil on the specimen support. The method simplifies cryo-EM specimen preparation, offering improved control over ice thickness and particle orientation, to help streamline and accelerate structure determination.

Cryo-EM has revolutionised structural biology, enabling larger structures, faster structure determination of complex specimens and in some cases true atomic-resolution structures of biological macromolecules[1,2]. This "resolution revolution" was enabled by transformative advances in microscope hardware, such as direct electron detectors, microscope automation and image processing developments[3–5]. These breakthroughs have established cryo-EM as a leading technique for exploring the structures of macromolecules and their complexes in biology, drug discovery and beyond[6].

Despite these technological developments, preparing high-quality cryo-EM grids remains a challenge. Because of the strong interaction of electrons with matter, the molecules of interest must be presented in vacuum and in a thin film of vitrified ice, to enable imaging by electrons and to avoid crystalline ice[7]. Currently the common sample preparation method creates the necessary thin films by adding the specimen to carbon or gold foil grids with small holes and blotting away the excess liquid, until free-standing thin films of the liquid are obtained. After that, the foil and films are frozen in liquid ethane to vitrify them. This method has not changed since its invention, despite efforts to automate it, and often involves a trial-and-error process[8]. It often requires the optimisation of numerous variables such as specimen concentration, buffer composition, and blotting and freezing parameters. Moreover, achieving precise control over the thickness of the aqueous specimen layer (~50 nm) prior to rapid freezing is challenging[9]. The resulting variability affects the quality of vitrified specimens and makes it necessary to screen many grids or leads to suboptimal data.

In recent years, alternative techniques for specimen deposition onto grids have emerged, offering improved control over specimen distribution and thickness[10]. Scribing methods, including pin printing and capillary writing, use sub-nanolitre volumes of specimen and provide reproducibility in ice thickness on grids by automating the deposition process[11,12]. Spraying techniques, on the other hand, show less control in reproducibility. However, they offer unique advantages, such as the ability to capture transient structural states due to the short interval between deposition and vitrification[13–15], which has also been exploited for time-resolved cryo-EM[16].

Apart from the reproducibility and variability issues, another major limitation arises from interactions between the specimen and surfaces, the foil and the air-water interfaces. Some specimens interact with the foil material (carbon in particular), leading to specimen loss[17]. Often, the majority of macromolecules in thin films generated by the aforementioned method are also adsorbed to the hydrophobic air-water interface, leading to two critical issues: partial or complete denaturation of macromolecules (more specimen loss) and a strong bias towards preferred orientations[18]. These problems stem from the high surface energy of water and the rapid Brownian diffusion of particles within thin films of ~50 nm thickness, leading to destructive encounters on a microsecond timescale[19–21].

[1]Medical Research Council (MRC) Laboratory of Molecular Biology, Cambridge, United Kingdom. [2]AstraZeneca, Cambridge, United Kingdom.
✉e-mail: yuezhang@mrc-lmb.cam.ac.uk; crusso@mrc-lmb.cam.ac.uk; jyl@mrc-lmb.cam.ac.uk

To address air-water interface-related issues, a variety of methods have been proposed. One involved modifying the grid surface with functionalised single-layer supports that selectively absorb proteins, minimising destructive interactions. Examples include chemically treated films (for example amorphous carbon or graphene) or grids with coatings designed to stabilise macromolecules[22–28]. Another approach employed sandwich structures, such as graphene or silicon nitride layers, which encapsulated the specimen on both sides and shielded it from the air-water interface[29,30]. Surface-active molecules such as detergents, lipids, or biofilm-forming proteins have also been incorporated into cryo-EM sample preparation workflows[31–33]. These molecules formed protective layers that reduce surface energy, mimicking the natural environment of biological membranes. Other methods attempted to reduce interaction by vitrifying the specimen before particles can reach the air-water interface[21,34]. Electrospray techniques, which used charged particles to shield proteins of interest, had also shown promise in reducing problems[35]. All of these methods made specimen vitrification for cryo-EM more complex, and some require extensive instrumentation and/or grid modifications. Here, we introduce a method utilising free-standing surfactant-stabilised foam films to tackle some of the challenges with current cryo-EM sample preparation.

## Results

### Foam film vitrification for cryo-EM

Foam films consist of thin aqueous films covered by amphiphilic molecules on both sides, and provide a well-known and easily accessible avenue for the generation of the desired geometry. Based on these, we combined the target macromolecules with soluble amphiphilic, surface-active molecules (Fig. 1A). A thin liquid film was drawn out using a wire loop made of copper, allowing both surfaces of the film to be covered with surface-active molecules to support the film. The film would otherwise collapse rapidly due to the strong surface tension of water. Films within the loop became thinner over second to minute time scales because of water evaporation, marginal regeneration and gravitation, marginal regeneration is a process in thin liquid films where thinner regions are replenished by liquid from adjacent thicker regions near the edges (plateau borders), driven by surface tension and pressure differences[36]. The film thickness can either be measured with laser interferometry or estimated visually by observing colour bands that are caused by interference of white light, which depends on the distance between the film surfaces. Once a desired thickness was reached, the film was transferred onto an electron microscopy grid by swiftly contacting the thin film with the grid and then vitrifying in liquid ethane. The grid, held with tweezers, was bent ~15° to align most of its surface parallel to the foam film, ensuring intact transfer. The apparatus needed to perform this procedure included a copper loop enclosing a circle 5.8 mm in diameter and having a 0.15 mm wire thickness (Fig. 1B), along with a locking holder (Fig. 1C) for the specimen container that had been designed to minimise specimen volume requirements (Fig. 1D). The container was made from poly(methyl 2-methylpropenoate) (PMMA), comprised two separable parts for cleaning purposes and a lid to minimise evaporation. The assembled container held ~20 μL of specimen in its central slot, into which the loop was immersed (Fig. 1E). Relevant dimensions of the container assembly are shown in Fig. 1F. A 0.5 mm-wide top opening accommodated excess liquid during loop insertion, while the extra 0.15 mm opening in the middle supported the thicker twisted wire above the loop, ensuring full immersion.

### Foam film vitrification generates uniform ice thickness and is independent of grid wettability

Grids prepared with the foam film method displayed consistent ice thickness across both Quantifoil and UltrAuFoil grids and yielded more usable grid squares than blotting techniques (Fig. 1G, H)[37,38]. The ice thickness gradient on the grid corresponded to that of the liquid film in the loop due to gravity.

The wetting properties of the grid itself are critical for blot-based vitrification methods and could also be expected to influence how our free-standing macroscopic foam films attach to and behave on grids. To investigate this, a grid holder with a mask covering half of the grid was made, for partially glow-discharging grids (Fig. S1A). The setup enabled direct comparison of the effect on the same grid, removing variability between grids. Grid wettability was assessed by measuring the water droplet contact angle on a half-glow-discharged grid (Fig. S1B). The non-glow-discharged area showed a contact angle of 90°, indicative of hydrophobicity, while the glow-discharged area had a contact angle of 18°, indicating hydrophilicity. HexAuFoil grids were chosen for these experiments because of their flat surfaces since in contrast to other grid types, their grid bars are level with the foil. Half of each grid was glow-discharged, and horse spleen ferritin with 0.01% DDM was used for the test.

The atlas of a half-glow-discharged HexAuFoil grid is shown in Fig. 2A, with the area above the red dashed line glow-discharged and the area below untreated. At low magnifications, no obvious differences in ice distribution were observed between the two areas. Equally, at higher magnifications, when examining grid squares, foil holes, and micrographs within the holes, no clear distinctions were apparent for the effect of grid wettability (Fig. 2B-G). Differences in ice distribution on the grid depended on film thickness. In thinner ice areas (< 80 nm) (Fig. 2B, E, and S1C), the ice was evenly distributed across the grid square. In regions with thicker ice (>100 nm) (Fig. S1D), the liquid tended to accumulate in the centre of the grid square. When using Quantifoil and UltrAuFoil grids instead, most grid squares exhibited a more uneven ice topology, as shown in Fig. S1E.

### Surfactants compatible with foam film vitrification

Lowering surface tension is a key requirement for surfactants to support free-standing foam films. To characterise and discover surfactants compatible with the foam film vitrification method, we measured the surface tension of surfactant solutions at 18 °C at different concentrations. Given that foam film formation is a dynamic process, the maximum bubble pressure method was employed to measure dynamic surface tension[39]. Figure 3A reports the dynamic surface tension profiles of four commonly used surfactants (detergents and lipids): DDM, LMNG, 06:0 PC, and 12:0 PC. For comparison, literature values of the critical micelle concentration (CMC) have been indicated as filled dots in Fig. 3A.

DDM and 06:0 PC exhibited typical S-shaped curves, with surface tension reaching a minimum at their respective critical micelle concentrations. In contrast, 12:0 PC showed no measurable surface tension reduction, maintaining a value similar to that of water at concentrations 1000 times above its critical micelle concentration. LMNG behaved uniquely, its surface tension decreased beyond the critical micelle concentration but remained high (~50 mN m$^{-1}$), even at a concentration of 1%, compared to DDM and 06:0 PC. To further investigate LMNG, we measured the surface tension as a function of bubble lifetime. The results showed that surface tension did not reach a minimum even at the maximum bubble lifetime of 20,000 ms achievable by our instrument (Fig. 3B). Surface tension measurements for other surfactants, including FOM, CHAPSO, GDN, OG, 07:0 PC and E. coli polar lipid extract are provided in Figure S2. Based on these results we continued with DDM as the surfactant of choice because of its low surface tension at a low critical micelle concentration and compatibility with many biological systems.

### Measurement of foam film thickness

As a more quantitative alternative to observation by eye, foam film thinning dynamics and film thickness were assessed prior to vitrification by analysing interference patterns generated by a laser

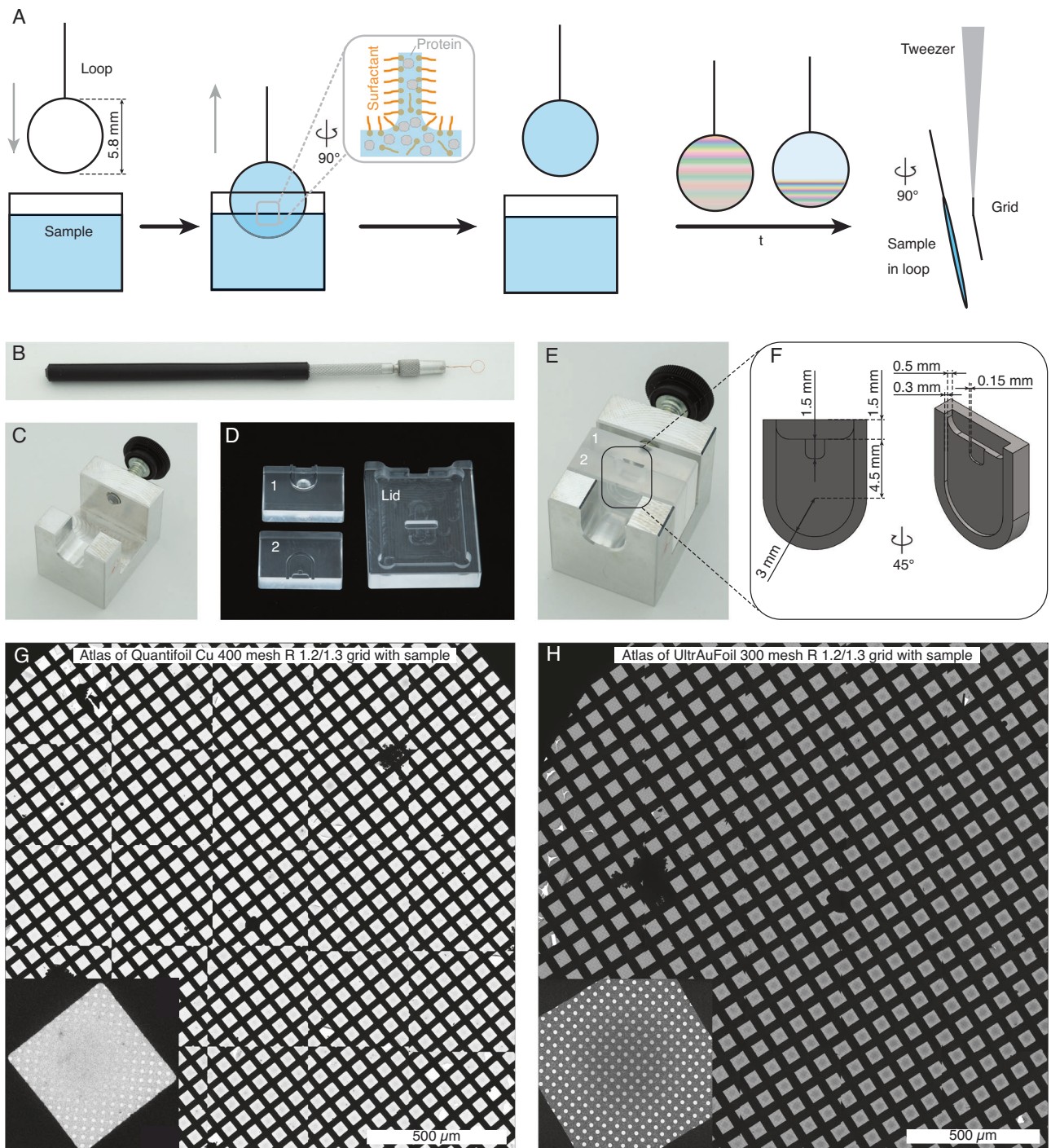

**Fig. 1 | Overview of the foam film cryo-EM vitrification method. A** A loop is immersed in a mixture of protein sample and surfactant. Upon withdrawal, a surfactant-coated foam film forms within the loop. Colour bands appear caused by the film surfaces due to light interference. Once the film reaches the desired thickness, by waiting time *t*, it is transferred onto a normal electron microscopy grid and plunge frozen. **B** The metal loop used in the experiment, with a diameter of 5.8 mm and a wire thickness of 0.15 mm. **C** Locking holder for assembling the sample container components (**D**). **E** The assembled container with a central slot for holding the sample and immersing the loop. **F** Dimensions of the slot, with an approximate volume of 20 μL. **G** Example overview composite micrograph (atlas) of a Quantifoil 400 mesh R 1.2/1.3 grid and (**H**) an UltrAufoil grid 300 mesh R 1.2/1.3 grid prepared using the foam film vitrification method, with enlarged grid square images in the lower right corner.

illuminating the film surfaces, similar to previously reported work by Sheludko[40] and Bergeron & Radke[41] (Fig. 4A). The experimental setup is shown in Fig. 4B. The interference patterns were recorded by a camera (Fig. 4C) and the average intensity at the centre of each frame over time was extracted. This allowed the film thinning speed, the timing of the film's disappearance (noted as $t_1$) and its final thickness to be determined (Fig. 4D).

Figure 4E reports the thinning speed over time for two specimen solutions: 0.01% DDM and 0.01% DDM with 1 g L$^{-1}$ EspB protein, measured under low (60%) and high (95%) humidity at 18 °C. The addition of EspB reduced the thinning speed under both humidity conditions. Films at higher humidity exhibited slower thinning speeds compared to those at lower humidity. When plotting the thinning speed as a function of film thickness, a divergence in thinning speed was

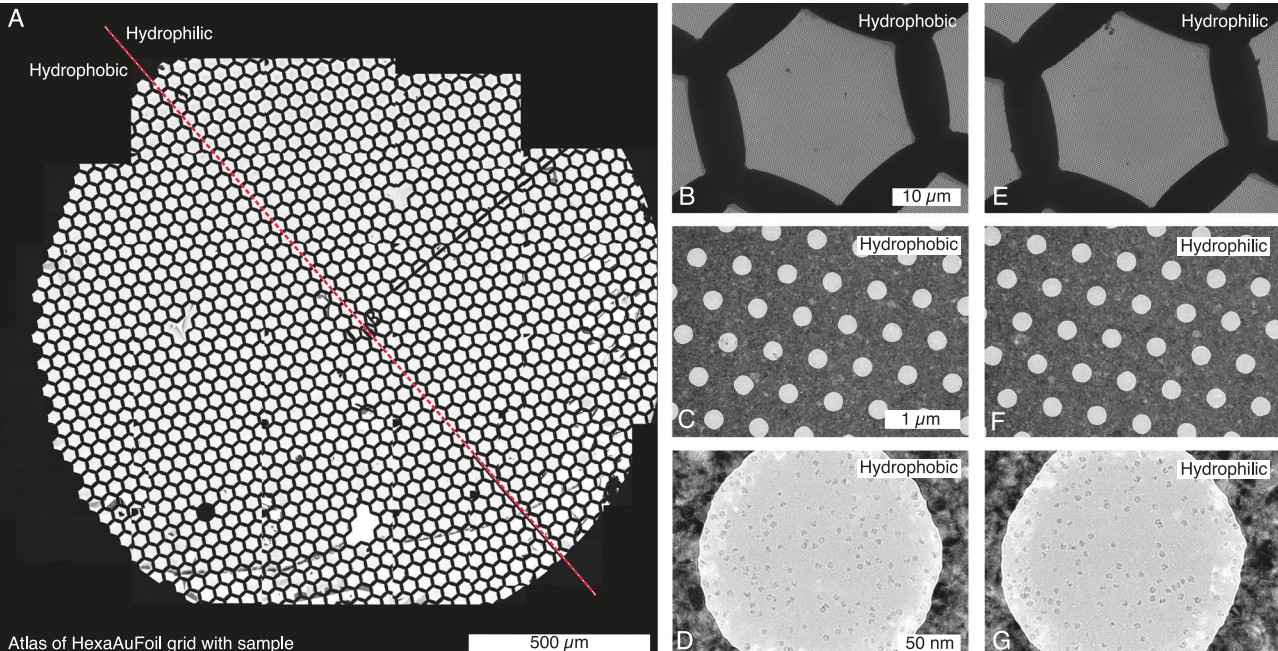

**Fig. 2 | Evaluating the effect of grid wettability on the foam film method.**
**A** Overview composite micrograph (atlas) of a half-glow-discharged HexAuFoil grid. The area above the red dashed line was glow discharged prior to the application of the foam film. Images from the hydrophobic portion of the grid include (**B**) a grid square, (**C**) foil holes, and (**D**) a horse spleen ferritin micrograph. Corresponding images from the hydrophilic portion include (**E**) a grid square, (**F**) foil holes, and (**G**) a horse spleen ferritin micrograph.

observed around thicknesses of 800–900 nm under different humidity conditions for these two specific solutions (Fig. 4F). At high humidity, the thinning speed of the film decreased faster as the film became thinner compared to that of low humidity condition. Figure S3 reports further foam film thinning speeds for solutions containing a variety of conditions.

### Correlation of laser- and electron microscope-based thickness measurements

To evaluate whether foam film thickness measurements prior to vitrification are predictive of the final ice thicknesses on grids after transfer and vitrification, we measured both foam film thickness and ice thickness on 12 grids and analysed the correlation. For this, first, in-hole ice thicknesses were determined using electron cryotomography (cryo-ET) on 88 holes, and a fitted line was calculated against pixel intensity measurements from the same holes in low magnification grid square images, showing a strong correlation with an $R^2$ value of 0.984 (Fig. 5A). The resulting plot agrees with previous studies on cryo-EM ice thickness measurements, with an inelastic mean free path of 293.4±8.1 nm[42,43]. Using this correlation, low magnification intensity measurements from ~50 to 100 holes per grid were then converted into estimated thicknesses for all 12 grids. This generated per-grid ice thickness distributions (Fig. 5B). A comparison between the foam film thicknesses, as measured by laser before the films were transferred to grids and vitrified, and the median ice thicknesses for the 12 grids as determined by EM after vitrification is shown in Fig. 5C. Re-plotting the foam film thickness versus ice thickness revealed a correlation that could be fitted by linear regression with an $R^2$ value of 0.824, indicating a good fit (Fig. 5D).

### Cryo-EM structure determinations using foam film vitrification

To assess the utility of the foam film vitrification method for cryo-EM, we prepared a range of macromolecular specimens with a variety of sizes and symmetries, and performed single-particle analysis on each. We determined seven structures, catalase, 30S ribosomal subunit, EspB, EspB 14-mers, 20S proteasomes, AcrB, and tau paired helical

filaments (PHFs) (Fig. 6A, E, I, M, P, S; Fig. S10), and assessed their resolutions by calculating the Fourier shell correlation (FSC) between two half-maps at 0.143[44]. The resulting resolutions ranged from 2.3 to 3.5 Å (Fig. S4–10). Three of these macromolecules, catalase, 30S ribosomal subunit, and EspB, exhibited fewer preferred orientations compared to specimens prepared without surfactant and using blotting in a Vitrobot. Using foam film vitrification, catalase had an orientation distribution efficiency of 0.8 as reported by cryoEF (Fig. 6B), compared to 0.15 for preparations using a Vitrobot without surfactant (Fig. 6C) and 0.21 with 0.01% DDM (Fig. 6D)[20]. For 30S ribosomes, the orientation distribution efficiency increased to 0.64 with foam film vitrification with 0.01% DDM, compared to 0.25 when using a Vitrobot without surfactant (Fig. 6F, G). When the 30S ribosome specimen was prepared using a Vitrobot with 0.01% DDM, the orientation distribution efficiency (0.65) was close to that of the foam film approach (Fig. 6H). For EspB, foam film vitrification with 0.01% DDM improved the orientation distribution efficiency to 0.77 (Fig. 6J), compared to 0.069 for Vitrobot-prepared specimens without surfactant and 0.04 for specimens prepared with 0.01% DDM using the Vitrobot (Fig. 6K, L). Using a different surfactant, 06:0 PC at 0.67%, with EspB, showed the same orientation distribution efficiency as that of using DDM at 0.77 (Fig. S7E). We obtained a different conformation, EspB 14-mers, when using 06:0 PC (Fig. 6M). For EspB 14-mers, 20S proteasomes and AcrB, specimens prepared via foam film vitrification exhibited orientation distribution efficiencies comparable to or better than those prepared with the Vitrobot (Fig. 6N, O, Q, R, T, U).

## Discussion

In this study, we presented a foam film vitrification method for single-particle cryo-EM sample preparation. We investigated foam film generation, the transfer of free-standing foam films onto grids, the influence of grid wettability, the choice of surfactant and demonstrate film thickness control. Using this approach, we determined high-quality cryo-EM structures, demonstrating wide applicability of the method (Fig. S4–11).

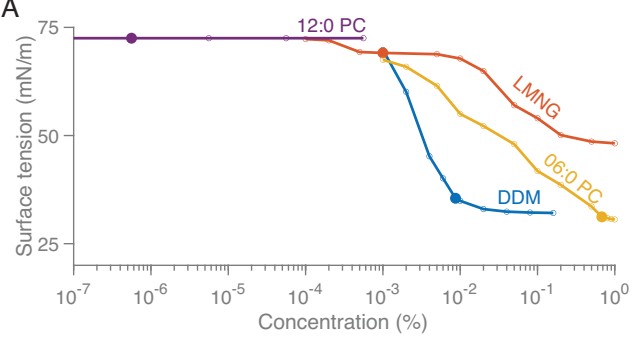

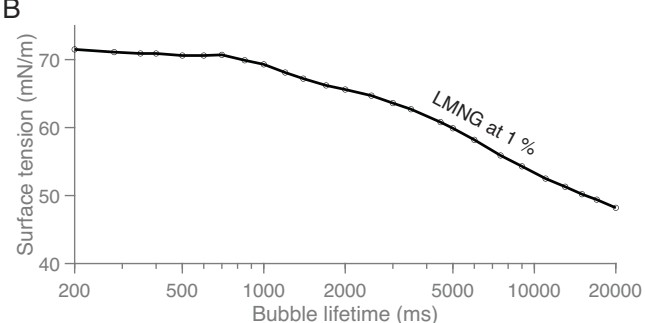

**Fig. 3 | Dynamic surface tension of different surfactants. A** Dynamic surface tension of DDM (blue), LMNG (red), 06:0 PC (yellow), and 12:0 PC (purple) measured at 18 °C using the maximum bubble pressure method. Filled dots indicate the critical micelle concentration of each surfactant. The bubble generation time was set to 20,000 ms in order to approximate equilibrium surface tension. **B** Surface tension of 1% LMNG at varying bubble generation times. The data used to generate the graphs are provided as a Source Data file.

To create a foam film, the target protein specimen is mixed with a surfactant. The loop used for generating the foam film has a diameter of 5.8 mm. This ensures adequate space in the centre of the loop for transferring the foam film to a standard EM grid of 3 mm diameter. The central region of the foam film exhibits less of a thickness gradient compared to the edges near the loop. In an automated setup, the loop size could be reduced, with both the loop and grid handled by robotic arms to position the grid precisely at the centre of the foam film. Using a thinner wire for the loop would allow modifications to the container design, reducing the required specimen volume to 10 μL or even 5 μL. The actual specimen consumption for each foam film was only tens of nanolitres, with unused liquid retrievable from the container. The container, made from PMMA, showed no obvious specimen adhesion issues for the tested proteins in this study. However, switching to other materials may further minimise protein adhesion with problematic and "sticky" specimens. During the film transfer to grids, bending the grid by ~15° ensured alignment parallel to the foam film, avoiding contact between the tweezers and the loop wire. However, excessive bending should be avoided, as it could lead to broken grid squares, due to the force applied when grids hit the surface of the liquid ethane (Fig. S1K).

Grids prepared using the foam film method exhibited variations in ice thickness both on the grid scale and within individual grid squares (Fig. S1). The grid scale ice thickness variations corresponded to the foam film thickness variation during drainage (thin in the middle and thick at the bottom). The variations observed within individual grid squares, such as liquid accumulations in the centre, did not appear to be related to grid wettability or surface topology. We suspected that redistribution of liquid occurs at the grid-square level once the film is applied, though the underlying mechanisms remained unclear. Further experiments will be required to investigate this phenomenon in detail.

Surfactants can be classified into anionic, cationic, zwitterionic and non-ionic types, based on their charge. In this study, we focussed on non-ionic surfactants since they are most compatible with proteins[45]. We also tested zwitterionic surfactants, such as phosphocholine derivatives with varying hydrophobic tail lengths, as zwitterionic lipids are abundant in cell membranes and are also often compatible with biological specimens.

Two key criteria governed surfactant selection, foamability and foam stability. Foamability ensures the formation of a foam film, while foam stability allows the film to persist long enough to thin to the desired thickness. Foamability is enhanced by reducing surface tension, which lowers the energy required to expand the liquid surface and create bubbles. Our measurements of dynamic surface tension confirmed that increasing surfactant concentration decreases surface tension, which reaches it minimum near the critical micelle concentration (Fig. 3A, Fig. S2).

Foam stability, which depends on surface elasticity, is as important as surface tension. For example, although the surface tension of CHAPSO solutions stabilises at 37 mN m$^{-1}$, higher than that of OG at 28 mN m$^{-1}$, foam films generated with CHAPSO were more stable. OG films often burst before thinning down to less than 100 nm, highlighting the importance of foam stability for this method. We speculated that the higher surface elasticity of CHAPSO, resulting from its molecular structure, contributed to this enhanced stability. Surface elasticity allows films to resist stresses such as those happening during drainage[46]. Factors influencing foam stability include surfactant structure, mobility at the air-water interface, intermolecular forces within the film and solution properties. Rheological properties, such as the ability to withstand dynamic expansion and contraction during foam generation and thinning, further influence (and complicate) film stability.

The thinning speed of vertical foam films depends on factors such as surfactant type, concentration, protein additives, and humidity. For thicker films (e.g. > 1000 nm), the thinning speeds were similar under both humidity conditions, as gravity-driven drainage dominates. However, as the film thins further, evaporation and marginal regeneration become more important[46–49]. High humidity extended film stability and lifetime, providing more time to transfer the film onto grids for vitrification (Fig. 4E, F).

Correlating laser-based foam film thickness measurements before transfer to a grid with the ice thickness after transfer and vitrification revealed some variability, leading to a good but not perfect correlation. We attributed this to the manual handling during the process as grids may not touch the same region of the film at the same angle each time and the timing between film transfer and vitrification may vary. Automating the process will increase consistency (Fig. 4D), although it will introduce complexity and cost.

High-resolution cryo-EM reconstructions after foam film vitrification were obtained for seven macromolecular specimens (Fig. S4–11). We showed that foam film vitrification was also suitable for membrane proteins and tau PHFs from tissues, since surfactants were already present in the samples[50,51]. For catalase, 30S ribosomes, and EspB, foam film vitrification improved orientation distributions. With the Vitrobot and blotting, catalase formed ribbon-like arrangements dominated by side views, while EspB specimens were dominated by top and bottom views (Fig. S12A, B, G, H). In contrast, foam film vitrification yielded particles with more random orientations (Fig. S12C, F, I).

We speculate that the improvements to particle orientation distributions were due to the surfactants forming monolayers at the air-water interface when the foam film was generated. These surfactant monolayers probably also helped distribute particles more evenly within the ice layer. For example, ferritin specimens vitrified with the foam film method showed uniform distribution of iron cores throughout both thick and thin ice, unlike Vitrobot-blotted specimens,

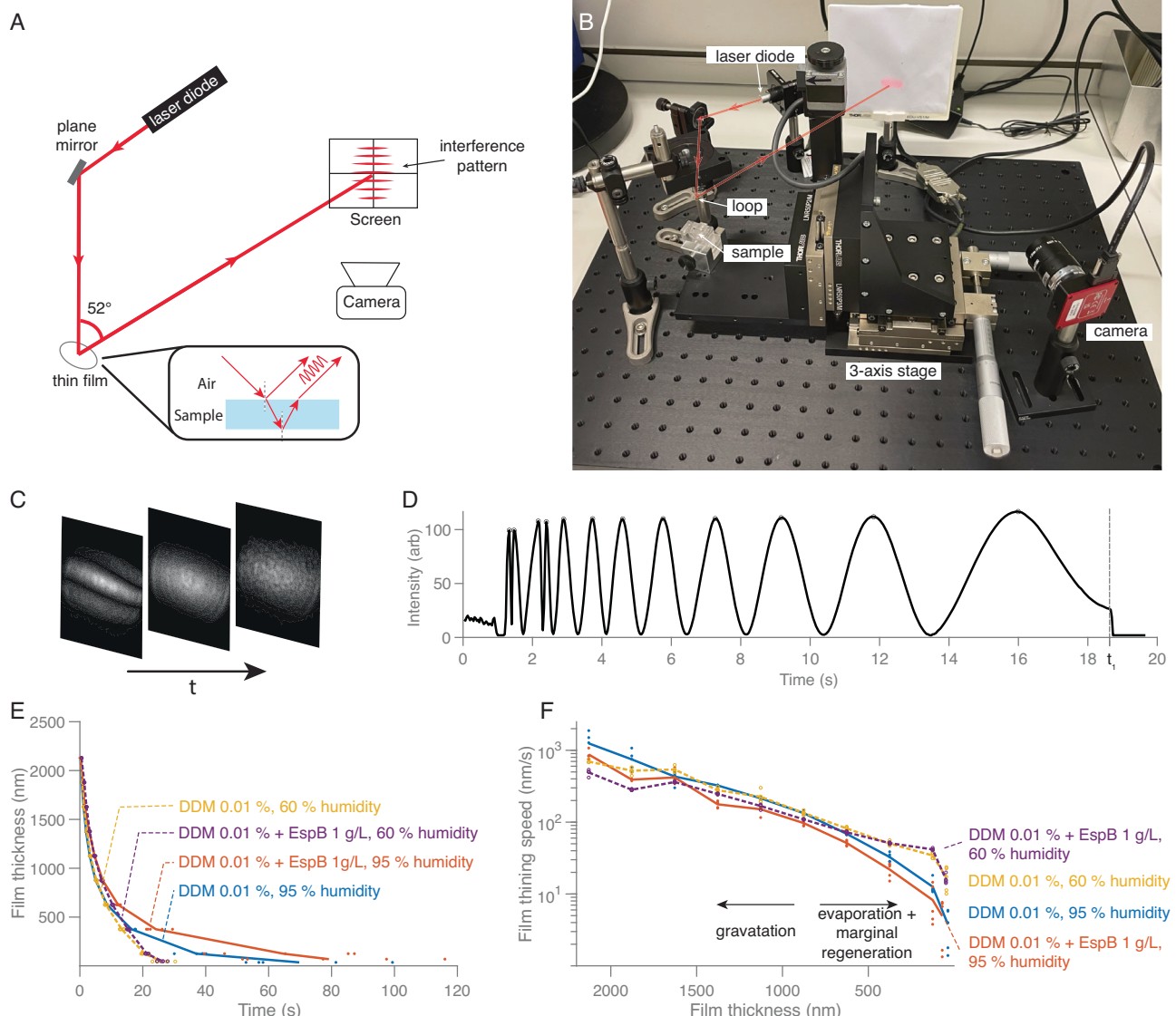

**Fig. 4 | Foam film thinning measurements under different humidity conditions.**
**A** Schematic of the experimental setup: a red laser (λ = 635 nm, 0.9 mW) is redirected by a plane mirror to illuminate the foam film inside the loop at an incident angle of 26°. The interference fringe created by reflections from both surfaces of the film is projected onto a screen and recorded by a digital camera. **B** Experimental setup, with the sample container on a 3-axis stage moving up and down in the z-direction at a constant speed of 1 cm s⁻¹ to enable the fixed loop to be immersed in the sample solution in the container. **C** Three snapshots of the interference pattern recorded by the camera as the foam film in the loop thins over time $t$. **D** Average intensity over a 30 × 30 pixel region at the middle of the images as a function of time; this particular foam film burst at time $t_i$. **E** Film thickness over time for 0.01% DDM, with and without 1 g L⁻¹ EspB, at 60% (dashed curves, yellow: DDM 0.01%, purple: DDM 0.01% + EspB 1 g L⁻¹) and 95% (solid curves, blue: DDM 0.01%, red: DDM 0.01% + EspB 1 g L⁻¹) humidity measured at 18 °C. Coloured dots represent individual measurements ($n = 5$) at specific thicknesses. **F** Film thinning speed versus thickness for the same conditions as (**E**). Coloured dots represent individual thinning speeds measurement at specific thicknesses. Supporting data for the graphs are available in the Source Data file.

where iron cores tended to accumulate at the surfaces (Fig. S13). Reconstructed tomograms of other specimens confirmed this (Movie S1 to S15).

Another benefit of the foam film vitrification method is improved spatial particle distribution within the foil holes of grids. Proteins such as 20S proteasomes and AcrB, which often adhere to carbon foil or concentrate near foil hole edges, even with the addition of surfactants, were more evenly distributed within the holes when prepared using foam film method (Fig. S14). We suggest this is because of the surfactants protecting the specimen from the grid surface as well.

Despite these advantages, the foam film method has some drawbacks. Reduced interactions with the air-water interface mean that higher macromolecular concentrations may be required, as previously noted by Vinothkumar and Henderson[52]. In practice, we

typically started with a concentration of 1 to 5 g/L, depending on the sizes of the macromolecules, and adjusted this based on the distribution observed in micrographs. As the foam film thins also due to evaporation, surfactant concentrations increase, sometimes above critical micelle concentrations, leading to micelle formation. These micelles (if present), together with surfactant monolayers on the film surfaces, contribute to unwanted background signals in micrographs, lowering the signal-to-noise ratio. We think this might be the reason that we were unsuccessful in solving the structure of haemoglobin (64 kDa), despite particles being visible in the micrographs (Fig. S15). Evaporation-driven thinning may also alter the pH and ionic strength of the specimen solution, introducing possibly unwanted changes[19]. For proteins with very high air-water interface affinity through avidity, such as for non-twisting FtsZ filaments, foam

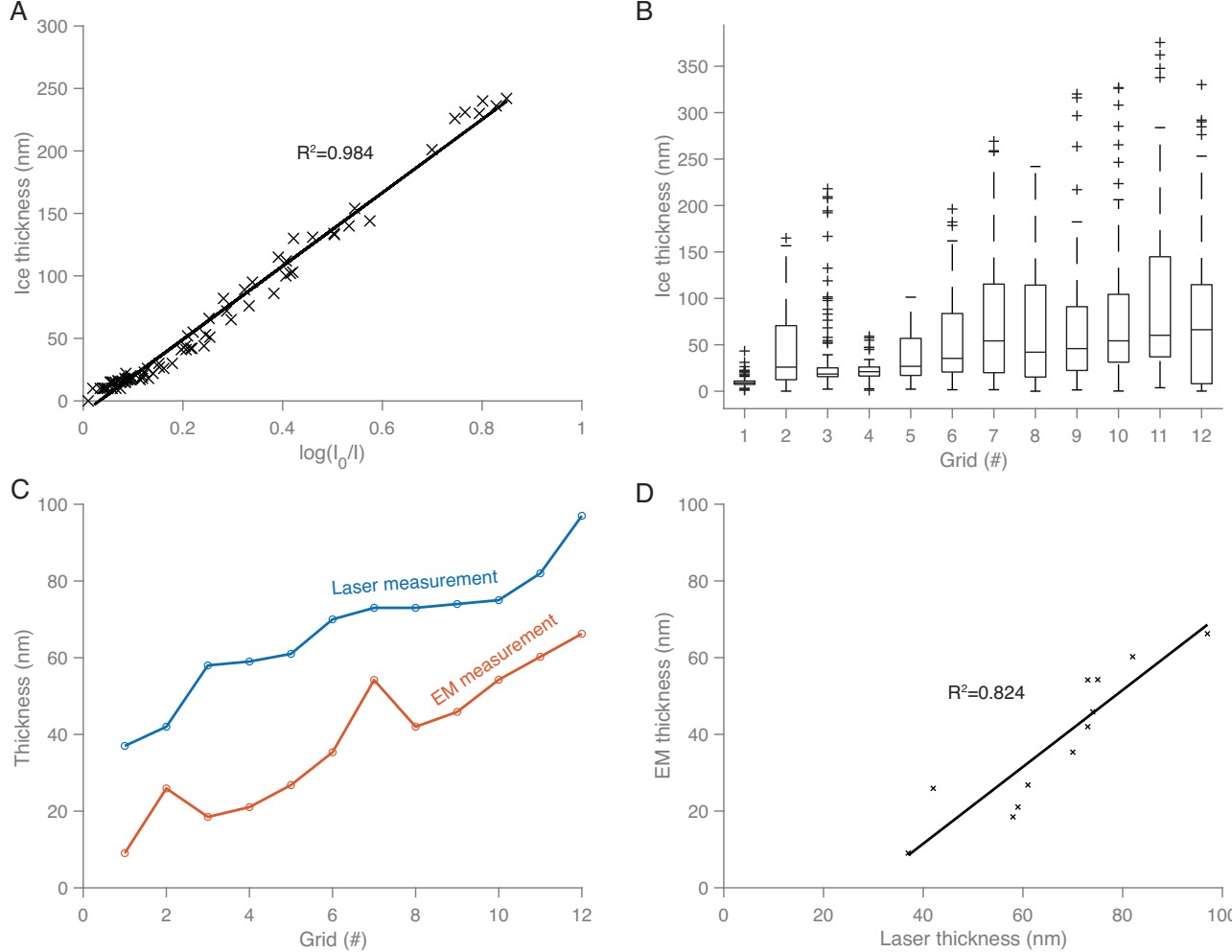

**Fig. 5 | Correlation between laser-measured foam film thickness inside the loop and ice thickness measured by electron microscopy after grid application and vitrification. A** Fitted line for ice thickness estimation by electron microscopy: normalised mean grayscale values of hole images (measured at $500 \times$ magnification) are plotted against ice thickness obtained by electron tomography ($n = 90$). $I_0$ is the intensity of an empty area, and $I$ is the measured intensity of the hole images. A fitted line is plotted, with a coefficient of determination $R^2$ value of 0.984. **B** Ice thickness distribution across $n = 12$ grids, with 91, 94, 95, 85, 74, 94, 86, 59, 87, 80, 84, and 74-hole measurements from grid 1 to 12, following the observed ice thickness gradient. The centre line of the box plot represents the median. The box edges indicate the 25th (lower quartile) and 75th (upper quartile) percentiles. Whiskers extend to the most extreme data points within 1.5 times the interquartile range from the quartiles, while points beyond this range are treated as outliers and shown individually. **C** Comparison of foam film thickness measured via laser (blue) and the median ice thickness measured by electron microscopy (red) for the same 12 grids. **D** Correlation between laser-measured foam film thickness and electron microscopy- measured ice thickness, with a fitted line showing a $R^2$ of 0.824. Data used to produce the graphs are supplied as a Source Data file.

film vitrification did not allow enough improvement of the orientation distribution to facilitate structure determination (Fig. S16). This result suggests that the DDM surfactant used may not form strong enough surface monolayers to prevent protein particles from interacting with the air–water interface. That also means that simply adding soluble detergents to specimen for cryo-EM will unlikely be able to solve the air-water interface problem for all samples when using other methods, such as blotting.

In summary, foam film vitrification simplifies grid preparation, reducing the need for glow-discharging, producing uniform ice thicknesses with improved particle distributions in many cases and enabling pre-vitrification control over the ice thickness in cryo-EM. The method also improved orientation distribution for some specimens. Future work will focus on employing stronger surfactants that often have much lower water solubility, such as lipids and fatty acids, and that are compatible with proteins[31,53]. These compounds are expected to produce much stronger surfactant monolayers that will help in cases where macromolecules interact very strongly with the air-water

interface or for small macromolecules where increased background is not an option. Completely coating foam film surfaces with water-insoluble surfactants is an engineering challenge, but techniques such as Langmuir-Blodgett films provide potential solutions[54,55]. These will broaden the applicability of foam film vitrification for cryo-EM, improving resolution and reliability for many, if not all macromolecular systems.

## Methods

### Grid wettability measurements
A customised grid holder with a mask covering half of the grid was designed and fabricated to hold the grid during glow-discharging (Fig. S1A). Grids were glow-discharged at 30 mA for varying durations: 60 s for Quantifoil and UltrAuFoil grids, and 180 s for HexAuFoil grids. After glow-discharging, the wettability of the grids was tested by applying water droplets to both the treated and untreated areas. The contact angle of water droplets was measured under a light microscope (Fig. S1B).

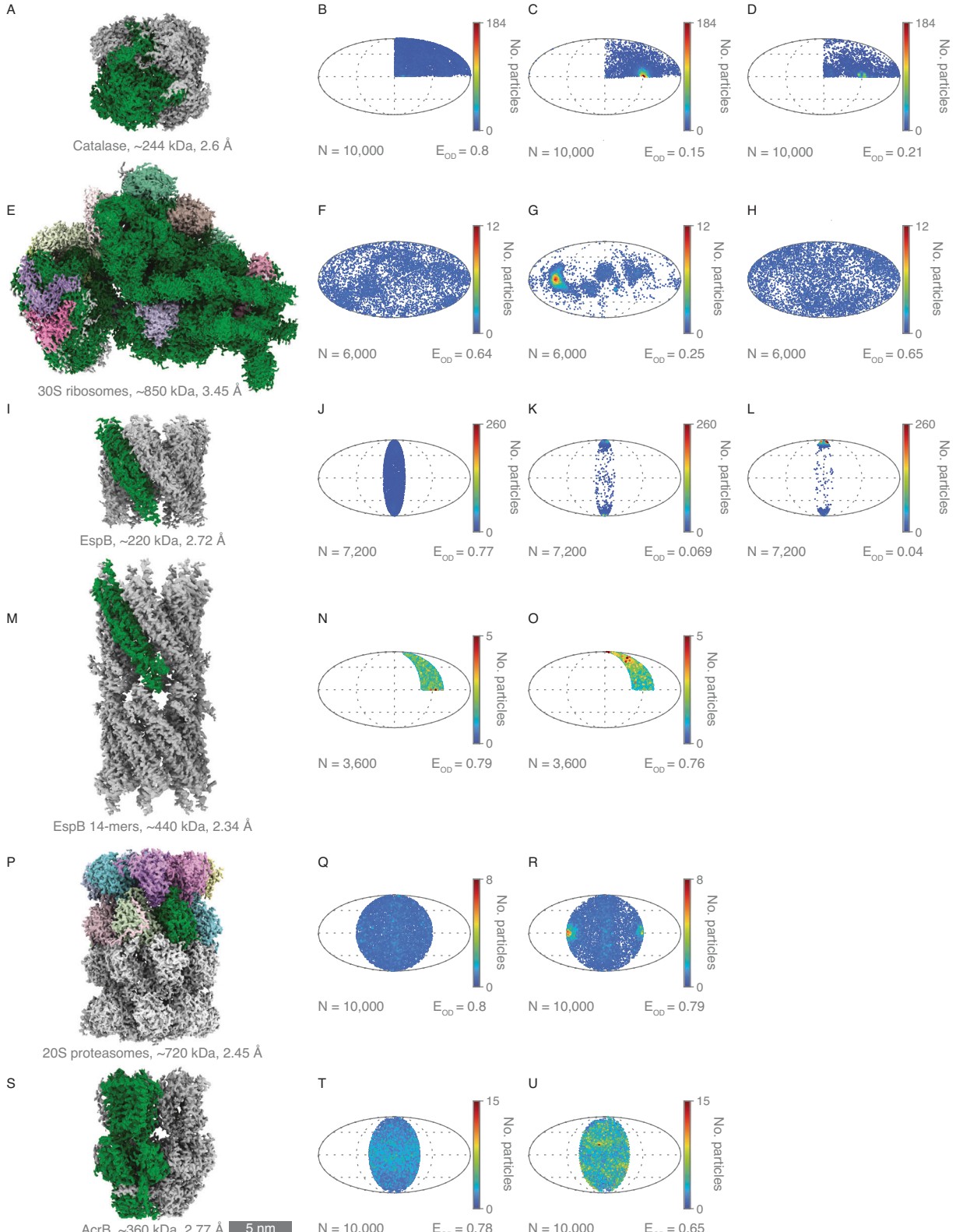

**Fig. 6 | Cryo-EM structures determined using the foam film vitrification method. A**, **E**, **I**, **M**, **P**, **S** 3D reconstructed cryo-EM maps for catalase, 30S ribosomes, EspB, EspB 14-mers, 20S proteasomes, and AcrB, respectively. **B**, **F**, **J**, **N**, **Q**, **T** Particle orientation distributions for each structure, with number of data points and orientation distribution efficiency indicated by cryoEF. **C**, **G**, **K**, **R** Particle orientation distributions of catalase, 30S ribosomes, EspB, and 20S proteasomes prepared using a Vitrobot without surfactant. **O**, **U** Particle orientation distributions of EspB 14-mers (with 0.67% 06:0 PC) and AcrB (with 0.03% DDM) prepared using a Vitrobot. **D**, **H**, **L** Particle orientation distributions for catalase, 30S ribosomes, and EspB prepared using a Vitrobot with 0.01% DDM. The data supporting the graphs are available in the Source Data file.

## Surface tension measurements

Surface tension of the surfactant solutions was measured using the maximum bubble pressure method with a SITA DynoTester+ (SITA Messtechnik GmbH, Germany). n-dodecyl-β-D-maltoside (DDM, Avanti, 850520), lauryl maltose neopentyl glycol (LMNG, Anatrace, NG310), n-octyl-β-D-glucopyranoside (OG, Melford Lab, N02007), glyco-diosgenin (GDN, Anatrace, GDN101), 3-[(3-cholamidopropyl)dimethylammonio]-2-hydroxy-1-propanesulfonate (CHAPSO, Sigma, C3649), (1H, 1H, 2H, 2H-perfluorooctyl)-β-D-maltopyranoside (FOM, Anatrace, O310F), 1,2-dihexanoyl-sn-glycero-3-phosphocholine (06:0 PC, Avanti, 850305), and 1,2-diheptanoyl-sn-glycero-3-phosphocholine (07:0 PC, Avanti, 850306) were each dissolved in Millipore water. 1,2-dilauroyl-sn-glycero-3-phosphocholine (12:0 PC, Avanti, 850335) and *E. coli* polar lipid extract (Avanti, 100600) were dissolved in chloroform, and then left at room temperature overnight to allow the solvent to evaporate, and water was added. For surface tension versus surfactant concentration experiments, a bubble lifetime of 20,000 ms was used to achieve a quasi-static surface tension value. For the dynamic surface tension versus bubble lifetime analysis of a 1% LMNG solution, the bubble lifetime was set to 200 ms and progressively increased to 20,000 ms. Each data point in the plots (Fig. 3 and S2) represents the average of 50 measurements.

## Foam film thickness and thinning measurements by laser interferometry

All surfactants used for the thinning measurements were dissolved in a buffer containing 50 mM Tris-HCl and 50 mM NaCl, adjusted to pH 7.5. The foam film thickness was measured by analysing the interference pattern generated when a laser illuminated the film surfaces (Fig. 4A). A 635 nm, 0.9 mW continuous-wave laser (PL202, Thorlabs, United States) was directed by a plane mirror and focussed by a lens to illuminate the centre of a foam film (focal spot size 1 mm) held within a loop at a 26° incident angle (Fig. 4B). The interference fringes, created by reflections from both film surfaces, were projected onto a screen and recorded by a digital camera at 30 fps with 540 × 720 pixels (CS165MU/M, Thorlabs, United States) (Fig. 4C). The specimen container, mounted on a 3-axis stage, was moved vertically at 1 cm s$^{-1}$ to immerse the fixed loop in the specimen solution. The average intensity over a 30 × 30 pixel region at the centre of each frame was plotted over time to determine the film thickness (Fig. 4D). Assuming the beam has an incident angle of 90°, the film thickness can be calculated as[40,41,56]:

$$d = \frac{\lambda}{2\pi n_l} \arcsin \left\{ \frac{\Delta}{1 + (4R/(1-R)^2)(1-\Delta)} \right\}^{1/2} \quad (1)$$

Here, $\lambda$ is the laser wavelength, $n_l$ is the refractive index of the liquid, approximated to be 1.33[56,57], and $\Delta = (I - I_{min})/(I_{max} - I_{min})$, $R = (n_l - 1)^2/(n_l + 1)^2$. The value $I$ is the instantaneous reflected intensity, while $I_{max}$ and $I_{min}$ correspond to the most recent interference maximum and minimum values. Equation (1) accounts for multiple reflections within the film.

To simplify, one could set $(4R/(1-R)^2)(1-\Delta) = 0$[40], Eq. (1) can be rewritten to:

$$d = \frac{\lambda}{2\pi n_l} \arcsin(\Delta^{1/2}) \quad (2)$$

When the incident angle $\theta_i$ is taken into account, the film thickness is:

$$d = \frac{\lambda}{2\pi n_l^* \cos \theta_r} \arcsin(\Delta^{1/2}) \quad (3)$$

where $\theta_r$ is the angle of refraction determined by Snell's law, $n_l \sin \theta_r = n^* \sin \theta_i$, and $n$ is the refractive index of air.

## Cryo-ET data collection and reconstruction

Cryo-ET data were collected on a Titan Krios 300 kV electron microscope (TFS) using automated acquisition via Tomo5 software. Tilt series were recorded at nominal magnifications of ×53,000 or ×81,000, with calibrated pixel sizes of 2.39 Å or 1.514 Å equipped with a 10 eV slit-width Selectris X energy filter and Falcon 4i camera, or at nominal magnifications of ×64,000 equipped with a Gatan K3 Summit camera and a GIF BioQuantum energy filter (slit width 20 eV), with calibrated pixel sizes of 1.38 Å. Data acquisition followed a dose-symmetric tilt scheme from −45° to +45° in 3° increments, with a defocus setting of −5 μm. Each tilt angle was recorded across 6 frames, resulting in a total fluence of 93 e$^-$ Å$^{-2}$ per tilt series. Motion correction and stage shift were corrected using MotionCor2[58], and tomogram alignment and reconstruction were performed in IMOD[59].

## Ice thickness measurements

For specimen preparations via the foam film method, Quantifoil 300 mesh R 1.2/1.3 grids (Quantifoil, Germany) were used. Horse spleen ferritin diluted to a concentration of 3 g L$^{-1}$ (Sigma, F4503) served as the specimen. Grids were held by tweezers, manually moved to touch the film, while the film thickness was measured by laser interferometry, and then plunge-frozen in liquid ethane. Foam films were prepared at room temperature (18 °C) and 60% humidity before vitrification.

In-hole ice thicknesses were calculated from reconstructed tomograms by measuring the distance between iron particles from ferritin on both film surfaces. Thicknesses measured from 88 tomograms were correlated with pixel intensity measurements from the corresponding holes in low magnification grid square micrographs, and a fitted line was plotted in Fig. 5A. The mean intensity of ice-containing holes ($I_i$) was measured at the centre of each hole, within a circular region of 0.8 μm in diameter, to avoid sharp thickness variations near the film edges. The difference in intensity over vacuum and over specimen is described by the Beer-Lambert law:

$$d_i = \lambda \log \frac{I_0}{I_i} \quad (4)$$

where $d_i$ represents the specimen thickness, $I_0$ is reference intensity over vacuum, $I_i$ is the intensity over ice, and $\lambda$ is the inelastic mean free path for electron scattering[60].

To ensure a statistically significant distribution of ice thicknesses, thicknesses were estimated from low magnification micrographs of grid squares across 12 grids. The calibration curve from Fig. 5A was then used to convert all intensity measurements (91, 94, 95, 85, 74, 94, 86, 59, 87, 80, 84, and 74-hole measurements from grid 1 to 12 across the thickness gradient) into estimated thicknesses, creating a per-grid ice thickness distribution for each measured grid (Fig. 5B).

## Preparation of biological specimens for cryo-EM single particle analysis

Quantifoil 300 mesh R1.2/1.3 grids were used for all specimens, except when noted otherwise.

Human erythrocyte catalase (Sigma, C3556) at a concentration of 2 g L$^{-1}$ was used. It was concentrated using Sartorius Vivaspin 500 Centrifugal Concentrators (30 kDa cut-off, VS0121) and the buffer exchanged to phosphate-buffered saline (PBS) at pH 7.4 to 3 g L$^{-1}$ or 6 g L$^{-1}$. For lower concentration than 2 g L$^{-1}$, sample was diluted with PBS. DDM was added at the final concentration of 0.2 mM (0.01%) before vitrification.

*E. coli* 30S ribosomes were purified as described in ref. 61, and concentrated to 3.4 g L$^{-1}$. DDM was added at the final concentration of 0.2 mM (0.01%) before vitrification. UltrAuFoil 300 mesh R1.2/1.3 grids were used.

EspB from *Mycobacterium tuberculosis* was purified as described previously[62]. The protein was used at 5 g L$^{-1}$. DDM at 0.2 mM (0.01%)

was added before vitrification. EspB 14-mers specimen were prepared by using EspB at 1.67 g L⁻¹, 06:0 PC at 15 mM (0.67%) was added before vitrification.

A human 20S proteasome core specimen was purchased from Enzo Life Sciences at 1 g L⁻¹ (BML-PW8720). The buffer was exchanged by dialysis at 4 °C for 2 h to 20 mM Tris-HCl, 1 mM EDTA, 1 mM DTT at pH 7.2. The concentration after buffer exchange was 0.7 g L⁻¹. DDM was added at the final concentration of 0.2 mM (0.01%) before vitrification.

AcrB from *E. coli* was purified as described previously[63]. The protein was used at 3 g L⁻¹ in buffer: 20 mM Tris-HCl, pH 7.5, 150 mM NaCl, and 0.6 mM (0.03%) DDM for preparing cryo-EM grids.

The tau paired helical filaments specimen from human brain tissues was purified as described previously[64]. UltrAuFoil 300 mesh R1.2/1.3 grids were used.

Lyophilised human haemoglobin (Sigma, H7379) was solubilised in 50 mM Tris-HCl buffer at pH 7.5 to a final concentration of 3 g L⁻¹. DDM was added at the final concentration of 0.2 mM (0.01%) before vitrification.

FtsZ from *E. coli* was purified as described[65]. The protein was used at 0.5 g L⁻¹ in a buffer solution of 50 mM Hepes-KOH, 100 mM potassium acetate, and 5 mM magnesium acetate, pH 7.7. Guanosine-5′-[(αβ)-methyleno]triphosphate (GMPCPP, Jena Bioscience, Jena, Germany) was added to 0.2 mM and incubated for 20 mins at 18 °C. DDM was added to a final concentration of 0.2 mM (0.01%) before vitrification.

For foam film vitrification, specimen preparation for single particle analysis was performed at 18 °C and 60% humidity with a manual plunger. The loop was held by hand, and the film thickness was visually estimated based on the colour bands visible (as described), before transferring the films to grids, also manually. For specimens prepared using the Vitrobot, all experiments were conducted with a Vitrobot Mark IV plunger set at 4 °C and 100% humidity.

### Single-particle cryo-EM data collection and data processing

Datasets for catalase, 30S ribosomes, paired helical filaments, haemoglobin, and FtsZ were acquired on a Titan Krios G3 microscope (TFS) equipped with a Falcon 4i camera, operating at 300 kV and controlled by EPU software. Imaging parameters included defocus ranges of −0.6 to −2.4 μm or −0.9 to −3 μm and pixel sizes of 0.824 Å, 0.67 Å, or 1.08 Å, respectively. Datasets for EspB, EspB 14-mers, and AcrB were collected on a Titan Krios G4 microscope (TFS) with a Falcon 4i camera and a Selectris X energy filter (slit width 10 eV), also operating at 300 kV with EPU software, using a defocus range of −0.6 to −2.4 μm and pixel sizes of 0.744 Å or 0.955 Å. The 20S proteasomes dataset was acquired on a Titan Krios G3 (TFS) with a Gatan K3 Summit camera and a GIF BioQuantum energy filter (slit width 20 eV), at 300 kV with defocus from −0.6 to −2.4 μm and a pixel size of 0.826 Å. Further details are provided in Table S1.

All cryo-EM datasets were processed using a consistent workflow. Raw detector data were converted into TIFF stacks (movies) with an appropriate electron dose per fraction (1 e⁻ Å⁻² per fraction). These movies were then imported into RELION 5.0 for processing[66]. Motion correction was applied using MotionCor2 with a 5 × 5 patch configuration[58]. Micrograph CTF parameters were estimated through an exhaustive search using CTFFIND-4.1[67]. Particles were manually picked or picked using 2D templates generated from manually picked classes or via CNN-based automatic picking in Topaz[68]. Particles were initially extracted with two-fold binning to accelerate alignment and classification, and initial models were created de novo from selected 2D classes. Particle sets were further refined by 3D classification, and the final sets were polished with multiple rounds of CTF refinement, which corrected for beam tilt, anisotropic magnification, higher-order aberrations, per-particle defocus, and per-micrograph astigmatism, followed by Bayesian polishing[69,70]. Final resolutions were determined by the Fourier shell correlation cutoff of 0.143 between two independently specimen half-maps[44].

### Reporting summary

Further information on research design is available in the Nature Portfolio Reporting Summary linked to this article.

## Data availability

The seven reconstructed maps generated in this study have been deposited in the EMDB database under accession code EMD-53038, EMD-53039, EMD-53040, EMD-53041, EMD-53042, EMD-53043, EMD-53044. The movies generated in this study have been deposited Figshare: https://doi.org/10.6084/m9.figshare.28566002.v1 Source data are provided with this paper.

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

## Acknowledgements
We thank A. Gijsbers from National Autonomous University of Mexico for the gift of the EspB specimen, X. Li for 30S ribosomes specimen, L. Gregory for the AcrB specimen, and D. Zhang for the tau paired helical filaments specimen (last three at the LMB). We thank N. Grant of the LMB VisLab for assisting with photography. We also thank the LMB EM facility, LMB Scientific Computing and the LMB Mechanical and Electronics workshops for technical assistance. We thank A. Gusach, F. van den Ent, D. Kureisaite-Ciziene, Y. Yang, C. Qi, S. Scheres, R. Henderson of the LMB and E. F. Garman from the University of Oxford for fruitful discussion. This project is supported through the Blue Sky research collaboration between AstraZeneca UK Limited and the UK Medical Research Council (project BSF2-08). This work was funded by Medical Research Council as part of UKRI (U105184326 to J. L. and MC_UP_120117 to C. J. R.).

## Author contributions
K.S., C.J.R. and J.L. conceived the study. Y.Z., B.N., C.J.R. and J.L. developed the methodology. Y.Z. and B.N. conducted the experiments. Y.Z. and J.L. wrote the manuscript with input from all authors.

## Competing interests
The authors declare no competing interests.
