## [Peer Review file · Nature Communications]

Foam film vitrification for cryo-EM

Corresponding Author: Dr Jan Löwe

Version 0:

Reviewer comments:

Reviewer #1

(Remarks to the Author)

Despite the recent great progress in cryo-EM single particle structural analysis of macromolecules towards high resolution and high throughput, sample vitrification on EM grid remains a major problem due to sample denaturation and/or preferred orientation by the interaction of sample molecules with the air-water interface. A variety of surfactants have been used to circumvent such problem to dates. The authors invented a rather simple tool to produce a thin form film of sample solution mixed with various surfactants and transfer it to holey EM grids for vitrification to make cryo-EM grids with desired ice thickness to study the benefit of this unique method for cryo-EM analysis. They carried out a thorough study on the physical properties of form film, such as thinning speed and surface tension, to select out DDM as the best surfactant and to determine its optimal concentration to be used in this method, and structural determination of seven different protein samples to investigate the effectiveness of the method over a conventional blotting method for vitrification. The results clearly demonstrate the great benefit of this method, producing cryoEM grids with distinctively less preferred orientations for many samples. This also suggests that the method will be quite effective for solving structures of denaturation sensitive samples by the interaction with air-water interface.

I really enjoyed reading the paper, following the logical design of the tools and experiments by the authors. I hope they will design and produce an automated device for form film formation, transfer and vitrification for the benefit of an ever-increasing number of cryo-EM users as soon as possible.

A few minor points for correction and consideration.

Line 297: “H” of “(Fig. 6G, H)” should not be here. “(Fig. 6H)” should be inserted at the end of the sentence ending in the middle of line 299.

Line 310: “(Fig. S10)” may be “(Fig. S4-10)”.

Line 327: “(thin in the middle and thick at the bottom)”?

Lines 328-330: The ice thickness within individual grid squares shows a clear tendency of thicker ice in around the center of square (Fig. S1G and I). Any speculation on this would be interesting.

Lines 353-354: It would be nice to give a brief explanation on “marginal regeneration” for general readers.

p. 27: Possibly “DDM” is missing at the end of Figure 6 legend.

Reviewer #2

(Remarks to the Author)

In their work, Zhang and colleagues address one of the major remaining challenges in cryo-EM – sample preparation. The thin foam film method they have developed offers numerous advantages compared to the traditional blotting method. It provides more uniform thin ice over the grid and improves particle distribution and orientations. This method could become a game changer for samples that exhibit strong preferred orientation, affinity for support films, and/or tendency to accumulate around the periphery of holes. The paper will be of great interest to the cryo-EM, structural biology, and drug development

audiences of Nature Communications and I enthusiastically recommend it for publication.

Minor comments

- What was the success rate of film transfers on grids? Were some grids partially covered?
- What was the measured mean free path from the fit in Fig. 5A?
- In Fig.4B the laser is labeled as "HeNe laser" while the laser used was a diode type.
- Caption of Fig. 6, (B, F, J, N, R, V) -> (B, F, J, N, Q, T)?
- line 141, the pixel size for the x81,000 magnification is larger than that for x64,000.
- line 156, "a circular region of 0.8 mm" should probably be 0.8 μm .

Radostin Danev

Reviewer #3

(Remarks to the Author)

Cryo-EM sample preparation is the major bottleneck for single particle analysis. In this manuscript, Zhang et al., used surfactant to stabilize foam film which was transferred to EM grid for vitrification. In such a way, they have shown that the DDM helped greatly to reduce preferred orientation problem in several soluble proteins. In addition, the thickness of ice could be controlled by the thickness of film. The distribution of proteins was improved using this method. Using this method, the structures of several proteins was solved at high resolutions. I have some major concerns here,

1. The thickness of ice is very important. The major advantage of this method is that the thickness of ice can be very well controlled by the thickness of film which is decided by the humidity and the time. As shown in Fig.4E, when the thickness of the film went below $\sim 600\text{nm}$, the decreasing of thickness was very much depending on humidity, which indicating that evaporation plays an important role here. Therefore, for example, from 600nm to 60nm , the concentration of surfactant is 10 times increased. Such an increasing of concentration would greatly reduce the contrast of cryo-EM image, which is harmful for structure determination, especially for proteins with small molecule weight ($<100\text{kD}$). The decreasing of contrast can be obviously visualized in supplemental figures. Indeed, for protein with relatively large molecule weight, tilting the stage is a simple way to deal with preferred orientation. For small proteins, the contrast of image at high tilted angle is not good enough for alignment. Therefore, preferred orientation problem is mostly remained on small proteins. The fact that this method is not good for small proteins makes it poor applicable.

2. A lot of cases (published paper) have shown that by adding 0.01% DDM, preferred orientation problem can be very well solved. However, in this study, for Catalase and EspB, adding 0.01% DDM did not help much here. In contrast, foam film did help. This is a very interesting phenomenon. Can you explain it? Is it because that the concentration of DDM in foam film method is indeed much more than 0.01%? Or I can see in Fig.S12H that, the thickness of DDM sample is very thin. Can you find some regions with thicker ice to collect data of EspB with 0.01% DDM? In the region with thicker ice, I guess that more DDM molecules are available to occupy the AWIs.

I have some minor points,

1. In the apoferritin sample prepared by foam film, was ferritin particles adsorbed by AWIs? If they were not, the distance between iron particles may not represent the thickness of ice.
2. You used different concentrations for different proteins. How to make the decision of what concentration to use?
3. When looking at Fig. 4F, it is hard for me to get the conclusion that the addition of EspB reduced the thinning speed under both humidity conditions (especially in the condition of 60%).
4. As a important parameter, molecular weight of each protein should be added in the Fig.6.

RESPONSE TO REVIEWERS

Reviewer #1 (Remarks to the Author):

Despite the recent great progress in cryo-EM single particle structural analysis of macromolecules towards high resolution and high throughput, sample vitrification on EM grid remains a major problem due to sample denaturation and/or preferred orientation by the interaction of sample molecules with the air-water interface. A variety of surfactants have been used to circumvent such problem to dates. The authors invented a rather simple tool to produce a thin form film of sample solution mixed with various surfactants and transfer it to holey EM grids for vitrification to make cryo-EM grids with desired ice thickness to study the benefit of this unique method for cryo-EM analysis. They carried out a thorough study on the physical properties of form film, such as thinning speed and surface tension, to select out DDM as the best surfactant and to determine its optimal concentration to be used in this method, and structural determination of seven different protein samples to investigate the effectiveness of the method over a conventional blotting method for vitrification. The results clearly demonstrate the great benefit of this method, producing cryoEM grids with distinctively less preferred orientations for many samples. This also suggests that the method will be quite effective for solving structures of denaturation sensitive samples by the interaction with air-water interface.

I really enjoyed reading the paper, following the logical design of the tools and experiments by the authors. I hope they will design and produce an automated device for form film formation, transfer and vitrification for the benefit of an ever-increasing number of cryo-EM users as soon as possible.

We would like to thank the Reviewer for their thoughtful and constructive comments.

A few minor points for correction and consideration.

Line 297: “H” of “(Fig. 6G, H)” should not be here. “(Fig. 6H)” should be inserted at the end of the sentence ending in the middle of line 299.

Thank you for pointing out these and the following mistakes – it is highly appreciated. Line 178 Fig. 6G, H is now Fig. 6F, G. Fig. 6H is now at the end of the sentence in line 180.

Line 310: “(Fig. S10)” may be “(Fig. S4-10)”.

Line 192: “(Fig. S10)” has been changed to “(Fig. S4-11)”.

Line 327: “(thin in the middle and thick at the bottom)”?

Line 208 now reads as: thin in the middle and thick at the bottom.

Lines 328-330: The ice thickness within individual grid squares shows a clear tendency of thicker ice in around the center of square (Fig. S1G and I). Any speculation on this would be interesting.

Indeed. This phenomenon does not appear to be related to surface wettability, as similar results were observed on both plasma-treated and untreated grids. It also does not seem to be influenced by grid surface topology, since similar behaviour was seen on HexAufoil—characterised by a very flat surface—as well as on Quantifoil and UltrAufoil, which have raised grid bars relative to the grid squares. One possible explanation is that the liquid prefers to reside in the holey foil regions to minimise the surface-to-volume ratio, driven by surface tension. However, as the underlying mechanism remains unclear, we prefer to refrain from drawing definitive conclusions.

Lines 353-354: It would be nice to give a brief explanation on “marginal regeneration” for general readers.

A sentence has been added in line 79-81: “marginal regeneration is a process in thin liquid films where thinner regions are replenished by liquid from adjacent thicker regions near the edges (Plateau borders), driven by surface tension and pressure differences.”

p. 27: Possibly “DDM” is missing at the end of Figure 6 legend.

“DDM” has been added to the end of the Figure 6 legend, thank you.

Reviewer #2 (Remarks to the Author):

In their work, Zhang and colleagues address one of the major remaining challenges in cryo-EM – sample preparation. The thin foam film method they have developed offers numerous advantages compared to the traditional blotting method. It provides more uniform thin ice over the grid and improves particle distribution and orientations. This method could become a game changer for samples that exhibit strong preferred orientation, affinity for support films, and/or tendency to accumulate around the periphery of holes. The paper will be of great interest to the cryo-EM, structural biology, and drug development audiences of Nature Communications and I enthusiastically recommend it for publication.

We thank the reviewer for their positive and helpful comments.

Minor comments

- What was the success rate of film transfers on grids? Were some grids partially covered?

The success rate of film transfer onto grids strongly depends on the user's level of experience. For trained users, transfer rates can be more than 80%. The most common failure occurs when the film breaks before it is applied to the grid. Partial coverage of the grid can happen when the grid is not parallel to the film during transfer, for example when grids are prepared manually as described in the manuscript. However, even in such cases, a good number of grid squares are typically well covered and suitable for data collection. We thought originally that the transfer would be a problem, but it turned out not to be.

- What was the measured mean free path from the fit in Fig. 5A?

The measured mean free path from the fit in Fig. 5A was 293.4 ± 8.1 nm. It has been added to line 159 and reads as: "The resulting plot agrees with previous studies on cryo-EM ice thickness measurements with the inelastic mean free path at 293.4 ± 8.1 nm".

- In Fig.4B the laser is labeled as "HeNe laser" while the laser used was a diode type.

Thanks to the reviewer for pointing this out! The laser module used here is indeed laser diode. It is now corrected in Fig. 4B as a laser diode.

- Caption of Fig. 6, (B, F, J, N, R, V) -> (B, F, J, N, Q, T)?

The caption of Fig. 6 has been modified to Fig. 6, (B, F, J, N, Q, T).

- line 141, the pixel size for the x81,000 magnification is larger than that for x64,000.

We apologise for the lack of clarity and thank the reviewer for highlighting this point. The sentence (line 335-338) has now been revised to: "Tilt series were recorded at nominal magnifications of x53,000 or x81,000, with calibrated pixel sizes of 2.39 Å or 1.514 Å on a Falcon 4i camera after a Selectris X energy filter set to 10 eV slit width, or at a nominal magnification of x64,000 on a Gatan K3 Summit camera (calibrated pixel size of 1.38 Å), after a GIF BioQuantum energy filter with the slit width set to 20 eV."

- line 156, "a circular region of 0.8 mm" should probably be 0.8 μ m.

Indeed. Now In line 353, 0.8 mm is changed to 0.8 μ m.

Reviewer #3 (Remarks to the Author):

Cryo-EM sample preparation is the major bottleneck for single particle analysis. In this manuscript, Zhang et al., used surfactant to stabilize foam film which was transferred to EM grid for vitrification. In such a way, they have shown that the DDM helped greatly to reduce preferred orientation problem in several soluble proteins. In addition, the thickness of ice could be controlled by the thickness of film. The distribution of proteins was improved using this method. Using this method, the structures of several proteins was solved at high resolutions. I have some major concerns here,

1. The thickness of ice is very important. The major advantage of this method is that the thickness of ice can be very well controlled by the thickness of film which is decided by the humidity and the time. As shown in Fig.4E, when the thickness of the film went below ~600nm, the decreasing of thickness was very much depending on humidity, which indicating that evaporation plays an important role here. Therefore, for example, from 600nm to 60nm, the concentration of surfactant is 10 times increased. Such an increasing of concentration would greatly reduce the contrast of cryo-EM image, which is harmful for structure determination, especially for proteins with small molecule weight (<100kD). The decreasing of contrast can be obviously visualized in supplemental figures. Indeed, for protein with relatively large molecule weight, tilting the stage is a simple way to deal with preferred orientation. For small proteins, the contrast of image at high tilted angle is not good enough for alignment. Therefore, preferred orientation problem is mostly remained on small proteins. The fact that this method is not good for small proteins makes it poor applicable.

We thank the reviewer for considering our work so carefully.

The reviewer is correct that small proteins pose a challenge for the foam film vitrification method under the conditions we have tested, as is described in the manuscript for haemoglobin. While the concentration of surfactant can increase during thinning, this effect largely depends on the extent to which evaporation contributes to the process. It is possible to prepare samples under high humidity conditions to minimise evaporation. In such cases, thinning will still occur through other mechanisms, but slower (as is described).

In this study, we have only tested a limited number of surfactants. It is conceivable that a surfactant could be identified that provides both low surface tension and film stability at concentrations below its CMC, thereby minimising the formation of micelles and maintaining image contrast in cryo-EM.

Additionally, using an ultra-thin wire, on the order of a few hundred nanometres in diameter, for the loop and maintaining a slow withdrawal speed from the solution could yield foam films with a much thinner initial thickness. This could help limit the increase in surfactant concentration during evaporation-driven thinning. However, such a setup would require a highly controlled environment with minimal vibration and disturbance, which is challenging to achieve through manual sample preparation. An automated system could potentially overcome these limitations.

Alternatively, as is hinted at the end of the discussion, largely insoluble surfactants may also provide a solution but pose the additional challenge of how to cover both surfaces completely since the surfactant cannot simply be dissolved in the sample solution before vitrification.

Overall, this work introduces a new approach to cryo-EM sample preparation, and there remains room for further exploration and optimisation.

2. A lot of cases (published paper) have shown that by adding 0.01% DDM, preferred orientation problem can be very well solved. However, in this study, for Catalase and EspB, adding 0.01% DDM did not help much here. In contrast, foam film did help. This is a very interesting phenomenon. Can you explain it? Is it because that the concentration of DDM in foam film method is indeed much more than 0.01%? Or I can see in Fig.S12H that, the thickness of DDM sample is very thin. Can you find some regions with thicker ice to collect data of EspB with 0.01% DDM? In the region with thicker ice, I guess that more DDM molecules are available to occupy the AWIs. I have some minor points,

We speculate that the beneficial effect of DDM in foam film vitrification, but not for Vitrobot-facilitated vitrification, may be attributed to the following:

1. In the foam film method, surfactant coverage of the film surfaces is essential to lower surface tension and maintain film stability during thinning. So, surfactant coverage is a prerequisite of the film to form in the first place. To demonstrate this point: when only EspB protein is added at 5 g/L, without any surfactant, the surface tension remained too high to form a stable foam film. Therefore, the successful formation of a foam film with surfactant added indicates that the film surfaces are indeed protected by the surfactant. Additionally, as thinning progresses, evaporation likely increases the local surfactant concentration beyond its CMC.
2. When using the Vitrobot, the final surfactant concentration on the grid is possibly lower than its CMC. At the CMC, surfactant molecules preferentially reside at the air–water interface. However, during Vitrobot preparation, blotting removes most of the sample volume, including the surfactant enriched interface, allowing newly exposed air–water interfaces to form with limited surfactant coverage because the surfactant concentration is now much lower. This increases the likelihood of protein adsorption at these interfaces, which may lead to denaturation or preferred orientations.

For EspB samples prepared with 0.01% DDM using the Vitrobot, Fig. 6L shows the particle orientation distribution map of a dataset with a highly preferred orientation distribution, mainly top and bottom views. The dataset includes regions of both thick and thin ice. The occasional side views likely originate from thicker areas, but their occurrence is too limited to improve angular coverage with $E_{OD} = 0.04$.

1. In the apoferritin sample prepared by foam film, was ferritin particles adsorbed by AWIs? If they were not, the distance between iron particles may not represent the thickness of ice.

We believe that in foam film samples, not all particles are adsorbed to the air–water interfaces but are instead mostly randomly distributed within the ice layer (Fig. S13). Given a large-enough number of ferritin molecules, such a distribution still reliably indicates the film thickness through its boundaries.

2. You used different concentrations for different proteins. How to make the decision of what concentration to use?

Theoretically, the number of particles present in the ice can be estimated based on the size and concentration of the sample molecules, assuming that the concentration within the frozen specimen matches that in free solution, as discussed by Vinothkumar and Henderson (2016). However, in the foam film method, evaporation is one of the mechanisms of film thinning, which can increase the sample concentration within the film. In practice, we typically start with a protein concentration of 1 to 5 g/L, depending on the protein size, and then adjust this concentration based on particle distribution observed in micrographs.

We added this to line 262-263: “In practice, we typically started with a concentration of 1 to 5 g/L, depending on the sizes of the macromolecules, and adjusted this based on the distribution observed in micrographs.”

3. When looking at Fig. 4F, it is hard for me to get the conclusion that the addition of EspB reduced the thinning speed under both humidity conditions (especially in the condition of 60%).

The effect of EspB on reducing the thinning rate is subtle, particularly in Fig. 4F. However, in Fig. 4E, this effect is more apparent, as the samples containing EspB consistently thinning more slowly than those without EspB. At 60% humidity, the difference becomes less noticeable because evaporation dominates the thinning process, which is largely independent of sample composition.

4. As a important parameter, molecular weight of each protein should be added in the Fig.6.

Molecular weight of each protein has now been added to Fig. 6.